# THE INFORMATION-AUTOENCODING FAMILY: A LAGRANGIAN PERSPECTIVE ON LATENT VARIABLE GENERATIVE MODELING

## ABSTRACT

A variety of learning objectives have been recently proposed for training generative models. We show that many of them, including InfoGAN, ALI/BiGAN, ALICE, CycleGAN, VAE, $\beta$-VAE, adversarial autoencoders, AVB, and InfoVAE, are Lagrangian duals of the same primal optimization problem. This generalization reveals the implicit modeling trade-offs between flexibility and computational requirements being made by these models. Furthermore, we characterize the class of all objectives that can be optimized under certain computational constraints. Finally, we show how this new Lagrangian perspective can explain undesirable behavior of existing methods and provide new principled solutions.

## 1 INTRODUCTION

Deep generative models have been successfully utilized in a wide variety of tasks (Radford et al., 2015; Zhu et al., 2017; Yang et al., 2017; Li et al., 2017b). Prominent examples include Variational Autoencoders (VAE, Kingma and Welling (2013); Rezende et al. (2014)) with extensions such as $\beta$-VAE (Higgins et al., 2016), Adversarial Autoencoders (Makhzani et al., 2015), and InfoVAE (Zhao et al., 2017); adversarially trained models such as Generative Adversarial Networks (Goodfellow et al., 2014) and their extensions such as ALI/BiGAN (Dumoulin et al., 2016a; Donahue et al., 2016), InfoGAN (Chen et al., 2016a) and ALICE (Li et al., 2017a); hybrid objectives such as CycleGAN (Zhu et al., 2017), DiscoGAN (Kim et al., 2017) and AVB (Mescheder et al., 2017). All these models attempt to fit an empirical data distribution, but differ in the measure of similarity between distributions, whether or not they allow for efficient (amortized) inference, and the extent to which the latent variables should retain or discard information about the data. These methods also crucially differ in terms of the optimization techniques used, which can be likelihood-based or likelihood-free (Mohamed and Lakshminarayanan, 2016).

In this paper, we revisit the question of designing a training objective for a latent variable generative model. We formulate a constrained (variational) optimization problem that explicitly captures the various desired properties of a generative model, such as fit to the data distribution, tractable (amortized) inference, and lossy vs. lossless compression. Surprisingly, we show that all the previously mentioned models are different Lagrangian duals of the *same primal optimization problem*. They only differ in the value assigned to the Lagrange multipliers, corresponding to different prioritizations of the various (potentially conflicting) desiderata. In fact, by considering all possible Lagrangian duals, we obtain a strictly more general class of models, of which InfoGAN, ALI/BiGAN, ALICE, VAE, $\beta$-VAE, adversarial autoencoders, AVB, CycleGAN/DiscoGAN and InfoVAE are all special cases.

This general formulation reveals the modeling trade-offs being made between flexibility and the tractability of computing and optimizing the resulting objective. We show that each of the previously mentioned models correspond to a special choice of the Lagrange multipliers that 'cancels out' certain intractable terms from the training objective, so that a particular computational constraint is satisfied (e.g., it can be optimized likelihood-based or likelihood-free). In addition, under some mild assumptions, we are able to characterize *all* the training objectives that can be optimized under certain computational constraints.

This generalization also provides new insights into the properties of these models, which are implied by particular choices of the Lagrange multipliers. First, solving the dual problem requires optimizaton over the Lagrange multipliers (to obtain the best approximation to the primal). However, existing models use fixed values for the Lagrangian multipliers. We explain the consequences of this approximation, and demonstrate scenarios where this approach fails to solve the primal. Furthermore we show that when the primal is infeasible (e.g., due to insufficient modeling capacity), the choice of Lagrange multipliers prioritizes different constraints, and show that appropriate prioritization can lead to improved log-likelihood score.

The paper is organized as follows. In Section 2, we start from an axiomatic derivation of VAEs from minimum desiderata, and realize that most generative model objectives can be written as the Lagrangian dual of a constrained variational optimization problem. In Section 3, we consider the more general case in which the Lagrangian multipliers can be arbitrarily selected, and enumerate over all possible objectives under different computation constraints. In Section 4, we introduce two applications of this formulation that allows us to diagnose potential problems, compare different approaches and design new models.

## 2 THE INFORMATION-AUTOENCODING FAMILY

We consider a class of probabilistic generative models with two types of variables: observed 'data' variables (denoted $x \in \mathcal{X}$) and latent 'feature' variables (denoted $z \in \mathcal{Z}$). For ease of exposition, we describe the case of discrete variables, but the results apply to continuous variables as well[1]. The model family specifies a joint probability distribution $p_\theta(x, z)$ parameterized by $\theta$. We assume that the distribution can be factored as

$$p_\theta(x, z) = p(z)p_\theta(x|z)$$

where $p(z)$ is a given, fixed distribution. We assume that it is efficient to sample from $p(z)$ and evaluate $p(z)$ for any $z$. Likewise, we assume it is easy to evaluate and to sample from $p_\theta(x|z)$ for any $x$ and $z$. In general, $p_\theta(x)$ is intractable to compute but easy to sample from (by sampling $(x, z)$ from $p_\theta(x, z)$ and discarding $z$); $p_\theta(z|x)$, on the other hand, is generally both difficult to evaluate and to sample from.

Given samples from a data distribution $p_{\text{data}}(x)$, the learning goal is to find $\theta$ such that $p_\theta(x) \approx p_{\text{data}}(x)$. The hope is often to discover latent structure in the data through the latent variables $z$, evocatively called 'features'. Given a data point $x$, one would want to infer the corresponding latent variables using $p_\theta(z|x)$. However, as noted above, that is generally intractable to evaluate and sample from. We consider the case of a separate *amortized inference* distribution $q_\phi(z|x)$, and assume $q_\phi(z|x)$ is easy to evaluate and to sample from. This results in the following approximation (lower bound) to the intractable $p_\theta(x)$, known as the ELBO:

$$\log p_\theta(x) \geq \mathbb{E}_{p_{\text{data}}(x)q_\phi(z|x)}[\log p_\theta(x|z)] - \mathbb{E}_{p_{\text{data}}(x)}[D_{\text{KL}}(q_\phi(z|x)||p(z))] \tag{1}$$

where $D_{\text{KL}}$ denotes the divergence: $D_{\text{KL}}(p||q) = \mathbb{E}_p[\log p - \log q]$.

### 2.1 AXIOMATIC DERIVATION OF VARIATIONAL AUTOENCODERS

Instead of deriving a learning objective as an approximation to the intractable marginal likelihood $p_\theta(x)$, we start from a list of basic properties that any learned model should satisfy. Intuitively, under ideal conditions, the generative model should model the true data distribution, while the amortized inference model $q_\phi(z|x)$ should capture the true posterior $p_\theta(z|x)$. We formalize this in the following definition:

**Definition 1** (Consistency Condition). *A distribution $p_\theta(x, z)$ over $\mathcal{X} \times \mathcal{Z}$, a distribution $p_{\text{data}}(x)$ over $\mathcal{X}$, and a family of conditional (inference) distributions $\{q_\phi(z|x)\}_{x \in \mathcal{X}}$ are said to be **consistent** with each other if the following holds:*

- ***Correct Data Marginal:*** $\forall x, p_\theta(x) = p_{\text{data}}(x)$.

- ***Correct Feature Inference:*** $\forall x$ *such that* $p_{\text{data}}(x) \neq 0$, $q_\phi(z|x) = p_\theta(z|x)$.

---

[1] under mild technical assumptions, such as the existence of the relevant densities.

We can alternatively express the consistency requirement using the following definition (Dumoulin et al., 2016b; Donahue et al., 2016)

**Definition 2** (Joint Inference Distribution). *Given $p_{\text{data}}(x)$ over $\mathcal{X}$, and a family of conditional (inference) distributions $\{q_\phi(z|x)\}_{x \in \mathcal{X}}$ we define the **joint inference distribution** as*

$$q_\phi(x, z) = p_{\text{data}}(x) q_\phi(z|x)$$

*For compactness, we denote $q(x) = p_{\text{data}}(x)$.*

**Corollary 1.** $p_\theta(x, z)$, $p_{\text{data}}(x)$ and $\{q_\phi(z|x)\}_{x \in \mathcal{X}}$ *are **consistent** with each other if and only if*

$$q_\phi(x, z) = p_\theta(x, z) \tag{2}$$

By noticing that the roles played by $z$ and $x$ are symmetric, we also have the following equivalent definition of consistency:

**Definition 3** (Feature Marginal and Posterior). *Given the joint distribution $q_\phi(x, z)$ defined in Definition 2 we define the following*

$$q_\phi(z) = \sum_x q_\phi(x, z) \qquad q_\phi(x|z) = q_\phi(x, z)/q_\phi(z)$$

**Corollary 2.** $p_\theta(x, z)$, $p_{\text{data}}(x)$ and $\{q_\phi(z|x)\}_{x \in \mathcal{X}}$ *are **consistent** with each other if and only if*

- *Correct Feature Marginal: $\forall z, q_\phi(z) = p(z)$.*

- *Correct Data Inference: $\forall z$ such that $p(z) \neq 0$, $q_\phi(x|z) = p_\theta(x|z)$.*

## 2.2 A Constrained Optimization Formulation

It is not difficult to see that for a given data distribution $p_{\text{data}}(x)$, there can be multiple distinct choices of $\theta, \phi$ satisfying the stated consistency condition (Chen et al., 2016b; Zhao et al., 2017; Li et al., 2017a). A few examples are provided in the appendix. To avoid this problem, we need to specify a preference for which solution is desirable. That is, we consider the following general optimization problem [2]

$$\min_{\theta, \phi} \quad f(\theta, \phi) \qquad s.t. \qquad p, \, p_{data}, \, q \text{ are consistent with each other} \tag{3}$$

Any $f(\theta, \phi)$ can in principle be used. In this paper, we study a special but important case:

$$f(\theta, \phi) = \alpha_1 I_p(x; z) + \alpha_2 I_q(x; z) \tag{4}$$

where $I_p(x; z), I_q(x; z)$ are mutual information between $x$ and $z$ under distributions $p(x, z)$ and $q(x, z)$ respectively. The coefficients $\alpha_1, \alpha_2$ can be either positive or negative, to maximize or minimize mutual . Both cases could be desirable, depending on the goal. Models like InfoGAN (Chen et al., 2016a) explicitly maximize mutual information with respect to a group of latent variables to ensure that this group of variables is used. Information maximization is also useful to avoid the uninformative latent code problem (Chen et al., 2016b; Zhao et al., 2017), where latent features are not used when using certain training objectives. In other scenarios we may wish to minimize mutual information. For example, the information bottleneck (Tishby and Zaslavsky, 2015) approach minimizes mutual information between input and latent features under the constraint that the features can still predict some labels. Similar ideas are also widely used in compression (Shamir et al., 2010) where minimum information is retained while still being able to perform certain tasks.

## 2.3 Variational Autoencoding as Constrained Optimization

To solve the above optimization problem (3) constrained with the consistency condition we use the notion of a divergence between probability distributions.

**Definition 4.** *Let $\mathcal{P}(\mathcal{A})$ be the set of all probability distributions over a set $\mathcal{A}$. A functional $D : \mathcal{P}(\mathcal{A}) \times \mathcal{P}(\mathcal{A}) \to \mathbb{R}^+$ is called a **strict divergence** if $D(p\|q) = 0$ if and only if $p = q$.*

---

[2] Starting from this section we neglect the subscript $p_\theta$ and $q_\phi$ to simplify the notation.

Examples of strict divergences include Kullback-Leibler (KL) divergence, maximum-mean discrepancy (with a universal kernel, see Gretton et al. (2007)), Jensen-Shannon Divergence (Nowozin et al., 2016), etc. In particular, the reverse KL divergence is defined as $D_{\neg\mathrm{KL}}(p\|q) = D_{\mathrm{KL}}(q\|p)$.

Given strict divergences $D$, and $D'$, and using the equivalent formulation of consistency in Definition 1 and Corollaries 1 and 2, we can define the following optimization problems with one of three possible constraints[3]:

$$\min_{\theta,\phi} \quad \alpha_1 I_p(x;z) + \alpha_2 I_q(x;z)$$

$$s.t. \quad D(p(x,z)\|q(x,z)) = 0 \tag{5}$$

$$\text{or} \quad D(q(x)\|p(x)) = 0 \,,\; D'(q(z|x)\|p(z|x)) = 0 \qquad \forall x \in \mathcal{X} \tag{6}$$

$$\text{or} \quad D(p(z)\|q(z)) = 0 \,,\; D'(p(x|z)\|q(x|z)) = 0 \qquad \forall z \in \mathcal{Z} \tag{7}$$

It is easy to see that using any of the three constraints (5)(6)(7) is equivalent to the original optimization problem in Eq.(3). In fact, any superset of the constraints (5)(6)(7) also leads to an equivalent optimization problem (with redundancy).

## 2.4 Variational Autoencoding as Lagrangian Dual

A natural approach to solve constrained optimization problems is to consider their Lagrangian relaxation. For example, the Lagrangian corresponding to (5) would be $f(\theta,\phi) + \lambda D(p(x,z)\|q(x,z))$ where $\lambda$ is the Lagrange multiplier. More generally, we define a broad class of Lagrangian relaxations corresponding to the constraints in (5), (6), and (7) and their supersets. This leads to the following general class of training objectives, which we shall see encompasses a large number of existing probabilistic autoencoding models.

**Definition 5.** *The Information(Info)-Autoencoding family is the set of all objectives of the form*

$$\mathcal{L}(\boldsymbol{\lambda}, \alpha_1, \alpha_2) = \alpha_1 I_p(x;z) + \alpha_2 I_q(x;z) + \sum_{i=1}^{n} \lambda_i \mathfrak{D}_i \tag{8}$$

*where $\alpha_1, \alpha_2 \in \mathbb{R}$, $\lambda_i \in \mathbb{R}$, $\lambda_i > 0$ are Lagrangian parameters, and the following are satisfied:*

*1) For all $i = 1, \cdots, n$, each $\mathfrak{D}_i$ is one of the following*

$$D(p(x,z)\|q(x,z)), \; D(p(x)\|q(x)), \; D(p(z)\|q(z))$$
$$\mathbb{E}_{r(x)}[D(p(z|x)\|q(z|x))], \; \mathbb{E}_{t(z)}[D(p(x|z)\|q(x|z))]$$

*where $D$ is any strict divergence, and $r, t$ are any distributions supported on $\mathcal{X}$, $\mathcal{Z}$ respectively*

*2) There exists strict divergences $D, D'$, and distributions $r, t$ supported on $\mathcal{X}$, $\mathcal{Z}$ respectively, so that one of the following sets $C_1, C_2, C_3$ is a subset of $\{\mathfrak{D}_1, \cdots, \mathfrak{D}_n\}$*

$$C_1 = \{D(p(x,z)\|q(x,z))\} \tag{9}$$
$$C_2 = \{D(p(x)\|q(x)), \mathbb{E}_{r(x)}[D'(p(z|x)\|q(z|x))]\} \tag{10}$$
$$C_3 = \{D(p(z)\|q(z)), \mathbb{E}_{t(z)}[D'(p(x|z)\|q(x|z))]\} \tag{11}$$

*We call this family the KL Information-Autoencoding family if in the above definition all divergences $D$ are either KL or reverse KL, $r(x)$ is $p(x)$ or $q(x)$, and $t(z)$ is $p(z)$ or $q(z)$.*

In the above definition, $r(x)$ is itself a set of Lagrangian parameters, corresponding to the set of constraints $\{D(q(z|x)\|p(z|x)) \mid x \in \mathrm{support}(q(x))\}$. In this paper we assume $r(x)$ is either $p(x)$ or $q(x)$. Similarly, we assume $t(z)$ is either $p(z)$ or $q(z)$. We leave the study of more general choices for $r, t$ as future work.

In general we must maximize over the Lagrangian dual parameters $\boldsymbol{\lambda} = (\lambda_1, \cdots, \lambda_n)$ to find the tightest relaxation to the primal. However we shall show in the following section that many existing models simply pick specific (suboptimal) values of $\boldsymbol{\lambda}$. In Section 3, we will show that this is due to tractability reasons: the Lagrangian is easy to optimize (in a certain sense) only for some $\boldsymbol{\lambda}$.

---

[3]To avoid cluttering the notation, we assume $p(x,z)$ and $q(x,z)$ are supported over the entire space $\mathcal{X} \times \mathcal{Z}$.

## 2.5 RELATIONSHIP WITH EXISTING MODELS

The learning objectives used to train many existing latent variable models (such as ELBO, InfoGAN, etc.) can be rewritten in the Lagrangian form introduced in Definition 5. By rewriting them explicitly as in Eq.(8), we uncover the implicit tradeoffs being made (by choosing the Lagrange multipliers, i.e., pricing constraint violations), revealing similarities and symmetries between existing models along the way.

**1)** The original evidence lower bound (ELBO) in Eq.(1) can be rewritten into equivalent Lagrangian dual forms

$$\mathcal{L}_{\text{ELBO}} = \mathbb{E}_{q(z)}[D_{\text{KL}}(q(x|z)\|p(x|z))] + D_{\text{KL}}(q(z)\|p(z)) \tag{12}$$

$$= \mathbb{E}_{q(x)}[D_{\text{KL}}(q(z|x)\|p(z|x))] + D_{\text{KL}}(q(x)\|p(x)) \tag{13}$$

The second form (13) is well known (Kingma and Welling, 2013) and is traditionally the explanation why optimizing ELBO leads to consistency condition. The same conclusion could also be derived symmetrically from Eq.(12).

**2)** ELBO has been extended to include a $\beta$ scaling parameter (Higgins et al., 2016)

$$\mathcal{L}_{\beta-\text{ELBO}} = -\mathbb{E}_{q(x,z)}[\log p(x|z)] + \beta\mathbb{E}_{q(x)}[D_{\text{KL}}(q(z|x)\|p(z))] \tag{14}$$

This can be written as

$$\mathcal{L}_{\beta-\text{ELBO}} = (\beta-1)I_q(x;z) + \mathbb{E}_{q(z)}[D_{\text{KL}}(q(x|z)\|p(x|z))] + \beta D_{\text{KL}}(q(z)\|p(z)) \tag{15}$$

which is the Lagrangian for Eq.(11) choosing $\alpha_1 = 0, \alpha_2 = \beta - 1$. This shows clearly the information preference of the model. If $\beta > 1$ this objective is information minimizing; if $\beta < 1$ this objective is information maximizing; and neutral when $\beta = 0$ which is the original ELBO objective. This neutrality corresponds to the observations made in (Chen et al., 2016b; Zhao et al., 2017).

**3)** The InfoGAN objective in (Chen et al., 2016a)[4] can be converted to Lagrangian form for Eq.(10)

$$\mathcal{L}_{\text{InfoGAN}} = -I_p(x;z) + \mathbb{E}_{p(x)}[D_{\text{KL}}(p(z|x)\|q(z|x))] + D_{\text{JS}}(p(x)\|q(x))$$

where $\alpha_1 = -1$, $\alpha_2 = 0$, and the Jensen-Shannon divergence $D_{\text{JS}}$ is approximately optimized in a likelihood-free way with adversarial training. This equivalent formulation highlights the mutual information maximization property of InfoGAN. It also reveals that the InfoGAN objective encourages correct inference because of the $D_{\text{KL}}(p(z|x)\|q(z|x))$ term.

**4)** InfoVAE/AAE (Zhao et al., 2017; Makhzani et al., 2015) written in Lagrangian form becomes

$$\mathcal{L}_{\text{InfoVAE}} = -I_q(x;z) + \mathbb{E}_{q(z)}[D_{\text{KL}}(q(x|z)\|p(x|z))] + D(q(z)\|p(z))$$

where $D$ is any divergence that can be optimized with likelihood-free approaches. This reveals an elegant symmetry between InfoGAN and InfoVAE. Both maximize mutual information, except InfoGAN is the Lagrangian of constraints in Eq.(10) while InfoVAE is the Lagrangian of those in Eq.(11).

**5)** ALI/BiGAN corresponds to direct minimization of $D_{\text{JS}}(q(x,z)\|p(x,z))$, while ALICE (Li et al., 2017a) has the following Lagrangian form:

$$\mathcal{L}_{\text{ALICE}} = -I_q(x;z) + \mathbb{E}_{q(z)}[D_{\text{KL}}(q(x|z)\|p(x|z))] + D_{\text{JS}}(q(x,z)\|p(x,z))$$

This is similar to InfoVAE except that the final divergence is taken with respect to the joint pair $(x,z)$ rather than $z$ only.

**6)** CycleGAN/DiscoGAN (Zhu et al., 2017; Kim et al., 2017) can be written as

$$\mathcal{L}_{\text{CycleGAN}} = -I_q(x;z) - I_p(x;z) + \mathbb{E}_{q(z)}[D_{\text{KL}}(q(x|z)\|p(x|z))] + \mathbb{E}_{p(x)}[D_{\text{KL}}(p(z|x)\|q(z|x))]$$
$$+ D_{\text{JS}}(q(x)\|p(x)) + D_{\text{JS}}(q(z)\|p(z))$$

This reveals clearly that the model is maximizing mutual information, and encouraging correct inference in both direction by both matching $q(x|z)$ and $p(x|z)$ and matching $p(z|x)$ and $q(z|x)$.

---

[4]The original InfoGAN applies information maximization only to a subset of latent variables $z$.

## 3 ENUMERATION OF TRACTABLE FAMILIES

In the previous section, we introduced a very broad family of training objectives encompassing several widely used approaches. While all the objectives in the family are reasonable, the special cases we have identified have a very special property in common: they are (in a certain sense) 'easy' to optimize. This is a highly non-trivial property to have. In fact, mutual information and most divergences used in Definition 5 are generally *not* tractable to compute and/or optimize. Thus, the objectives will be 'easy' to optimize only if we carefully choose certain values of the Lagrange multipliers $\lambda_i$, so that certain terms 'cancel out'.

In this section, we propose a general framework to formally characterize *all training objectives in the family that satisfy certain tractability requirements*. We formalize tractability by defining the notion of a tractable "tool box", corresponding to a set of terms that we assume can be computed/optimized reasonably well, such as the ones used in ELBO (Kingma and Welling, 2013; Rezende et al., 2014), InfoGAN (Chen et al., 2016a), InfoVAE (Zhao et al., 2017), etc. Note that these objectives are generally not *provably* tractable to optimize, as they involve non-convexities, expectations over high-dimensional spaces, etc. There is, however, significant empirical evidence that they can be optimized reasonably well in practice, and we therefore group them based on the techniques employed.

The main technical challenge is dealing with all the possible simplifications and transformations that leave an objective invariant. For example, we need to account for objectives like (15), which appear to involve components that are difficult to estimate such as $I_q(x; z)$, but can actually be simplified to a more manageable form (14) that can be optimized in a likelihood-based way.We therefore define a set of transformations that leave an objective equivalent according to the following definition.

**Definition 6.** *An objective $\mathcal{L}$ is equivalent to $\mathcal{L}'$ when $\exists C$, so that for all parameters $\theta, \phi$, $\mathcal{L}(\theta, \phi) = \mathcal{L}'(\theta, \phi) + C$. We denote this as $\mathcal{L} \equiv \mathcal{L}'$.*

We are then able to analyze all the objectives that contain elements from a given "tool box", and all objectives derived from them through equivalent transformations.

### 3.1 LIKELIHOOD-BASED AND LIKELIHOOD-FREE COMPONENTS

We formalize the idea of "computation toolbox" by the following definition:

**Definition 7.** *For any divergence $D$, we define*

*(1) Likelihood-based terms as following set*

$$T_1 = \{\mathbb{E}_p[\log p(x|z)], \mathbb{E}_p[\log p(x, z)], \mathbb{E}_p[\log p(z)], \mathbb{E}_p[\log q(z|x)]$$
$$\mathbb{E}_q[\log p(x|z)], \mathbb{E}_q[\log p(x, z)], \mathbb{E}_q[\log p(z)], \mathbb{E}_q[\log q(z|x)]\}$$

*(2) Unary likelihood-free terms as the following set*

$$T_2 = \{D(q(x)\|p(x)), D(q(z)\|p(z))\}$$

*(3) Binary likelihood-free terms as the following set*

$$T_3 = \{D(q(x, z)\|p(x, z))\}$$

*Define an objective $\mathcal{L}$ as likelihood-based computable if there exists equivalent transformations $\mathcal{L}' \equiv \mathcal{L}$ so that $\mathcal{L}'$ only contains terms in $T_1$; unary likelihood-free computable if $\mathcal{L}'$ only contains terms in $T_1 \cup T_2$; binary likelihood-free computable if $\mathcal{L}'$ only contains terms in $T_1 \cup T_2 \cup T_3$.*

Consider the set $T_1$ of likelihood-based terms. Because we can directly compute likelihoods for $\log p(x|z)$, $\log p(x, z)$, $\log p(z)$ and $\log q(z|x)$, optimizing over an expectation over these terms can be achieved by back-propagation using path-wise derivative methods (the reparameterization trick) (Kingma and Welling, 2013; Schulman et al., 2015; Jang et al., 2016; Maddison et al., 2016), and by likelihood ratios when reparameterization is not applicable (Mnih and Gregor, 2014; Schulman et al., 2015).

Likelihood-free methods have been extensively studied in recent years. Examples include Jensen Shannon divergence (Goodfellow et al., 2014), f-divergences (Nowozin et al., 2016), Wasserstein

distance (Arjovsky et al., 2017; Gulrajani et al., 2017) and Maximum Mean Discrepancy (Gretton et al., 2007; Li et al., 2015). Even though such models are still difficult to train, significant improvements have been made in terms of stability and efficiency. Here, we distinguish between the (unary) case, where we consider a single group of variable at a time (either $x$ or $z$), and the more general binary case. The reason is that the binary case has been empirically shown to be more difficult to train, for example leading to inaccurate inference (Dumoulin et al., 2016b; Donahue et al., 2016).

## 3.2 ELEMENTARY TRANSFORMATIONS

Many elementary probabilistic identities can be used to rewrite an objective into another one which is equivalent according to Definition 6. An example is the chain rule identity

$$p(x, z) = p(x)p(z|x) = p(z)p(x|z)$$

In addition, terms that do not contain trainable parameters ($\phi, \theta$) can be freely added to the objective without affecting the optimization. Examples include $\mathbb{E}_p[\log p(z)]$ and $\mathbb{E}_q[\log q(x)]$. We formalize the above intuition in the following definition.

**Definition 8.** *Define the following set of transformations as the elementary transformations*

$$\mathbb{E}_*[\log p(x, z)] \equiv \mathbb{E}_*[\log p(x) + \log p(z|x)] \equiv \mathbb{E}_*[\log p(z) + \log p(x|z)]$$
$$\mathbb{E}_*[\log q(x, z)] \equiv \mathbb{E}_*[\log q(x) + \log q(z|x)] \equiv \mathbb{E}_*[\log q(z) + \log q(x|z)]$$
$$\mathbb{E}_q[\log q(x)] \equiv 0 \qquad \mathbb{E}_p[\log p(z)] \equiv 0$$

*where $\mathbb{E}_*$ indicates that the equivalence holds for both $\mathbb{E}_p$ and $\mathbb{E}_q$.*

Note that this set does not contain all possible transformations, such as importance sampling

$$\mathbb{E}_q[\log p(x)] \equiv \mathbb{E}_{q(x)}\left[\log \mathbb{E}_{q(z|x)}\left[\frac{p(x, z)}{q(z|x)}\right]\right]$$

We leave extensions to more general transformations to future work. Nevertheless, we are now in a position to study *all* the objectives that can be constructed using the building blocks from Definition 7, and any objective derived by applying elementary transformations. From now on, we restrict the definition of tractable family in Definition 7 to equivalence by elementary transformation only.

## 3.3 ENUMERATION OF KL INFORMATION-AUTOENCODING FAMILY

In the previous sections we have formally defined a grammar for tractable models. In this section, we will use these definitions to find all possible objectives that can be optimized under given tractability assumptions. We shall write the largest family under each tractability constraint in two forms: a Lagrangian dual form, and an equivalent tractable form that only contains terms satisfying assumed computability constraints.

In this section we focus on the KL Information-Autoencoding family defined in Definition 5 where all divergences in the constraints are either $D_{\mathrm{KL}}$ or $D_{\neg\mathrm{KL}}$.[5] We discuss KL divergence because it is the most interesting case to study under elementary transformations. KL divergence is a sum of expectations over log probabilities $D_{\mathrm{KL}}(p\|q) = \mathbb{E}_p[\log p - \log q]$ while the elementary transformations are all linear operations on log probabilities. We will discuss generalization to other divergences in the next section.

For likelihood based computable families, we show that variational mutual information maximization (Barber and Agakov, 2003) and $\beta$-ELBO (Higgins et al., 2016) are the only possible families (up to equivalences). Based on the set of transformations we considered, there do not exist other objectives satisfying the stated computational constraints. This is formally given by the following theorem.

**Theorem 1.** *Define the following objectives*

---

[5]To avoid cluttering the notation, if a $D_{\mathrm{KL}}$ term has free variables such as $z$ in $D_{\mathrm{KL}}(q(x|z)\|p(x|z))$, then we take the expectation for that free variable with respect to $p$ when the first distribution of the KL divergence is $D_{\mathrm{KL}}(p\|\cdot)$, and with respect to $q$ otherwise, e.g. $\mathbb{E}_{q(z)}[D_{\mathrm{KL}}(q(x|z)\|p(x|z))]$, $\mathbb{E}_{p(z)}[D_{\mathrm{KL}}(p(x|z)\|q(x|z))]$.

*1) Variational mutual information maximization*

$$\mathcal{L}_{\text{VMI}} = -I_p(x; z) + D_{\text{KL}}(p(z|x)\|q(z|x)) = \mathbb{E}_p[\log q(z|x)]$$

*2) $\beta$-ELBO, where $\beta$ is any real number*

$$\begin{aligned}\mathcal{L}_{\beta-\text{VAE}} &= (\beta - 1)I_q(x, z) + D_{\text{KL}}(q(x|z)\|p(x|z)) + \beta D_{\text{KL}}(q(z)\|p(z)) \\ &= -\mathbb{E}_q[\log p(x|z)] + \beta \mathbb{E}_q[\log q(z|x)] - \beta \mathbb{E}_q[\log p(z)]\end{aligned}$$

*Then any likelihood based computable objective of the KL Info-Autoencoding family is equivalent by elementary transformation to a linear combination of $\mathcal{L}_{\beta-\text{ELBO}}$ and $\mathcal{L}_{\text{VMI}}$.*

We formally prove this theorem in the appendix and give a brief sketch here. This theorem is proved by demonstrating that a set of terms such as $\mathbb{E}_q[\log p(x)]$ can be selected, so that each objective of the KL Information-Autoencoding family is a linear combination of these basic terms. Treating the set of basic terms as basis for a vector space (over functions), each training objective can be represented as a vector in this space. Each family of objectives can be represented as a subset and in some cases as a linear subspace. The key technical difficulty is to show that the following family of objectives correspond to the same linear subspace:

1. Linear combinations of $\mathcal{L}_{\beta-\text{ELBO}}$ and $\mathcal{L}_{\text{VMI}}$, and their elementary transformations.

2. Likelihood-based objectives (as in Definition 7), and their elementary transformations.

For unary likelihood free computable objectives, we show that a slight generalization of InfoGAN and InfoVAE are the only possible families. This is formally given by the following theorem.

**Theorem 2.** *Define the following objectives*

*3) KL InfoGAN, where $\lambda_1, \lambda_2$ are any real number*

$$\begin{aligned}\mathcal{L}_{\text{InfoGAN}} &= -I_p(x; z) + D_{\text{KL}}(p(z|x)\|q(z|x)) + \lambda_1 D_{\text{KL}}(p(x)\|q(x)) + \lambda_2 D_{\text{KL}}(q(x)\|p(x)) \\ &= -\mathbb{E}_p[\log q(z|x)] + \lambda_1 D_{\text{KL}}(p(x)\|q(x)) + \lambda_2 D_{\text{KL}}(q(x)\|p(x))\end{aligned}$$

*4) KL InfoVAE, where $\alpha, \lambda_1, \lambda_2$ are any real number*

$$\begin{aligned}\mathcal{L}_{\text{InfoVAE}} &= \alpha I_q(x; z) + D_{\text{KL}}(q(x|z)\|p(x|z)) + \lambda_1 D_{\text{KL}}(q(z)\|p(z)) + \lambda_2 D_{\text{KL}}(p(z)\|q(z)) \\ &= -\mathbb{E}_q[\log p(x|z)] + (\alpha + 1)\mathbb{E}_q[\log q(z|x)] - (\alpha + 1)\mathbb{E}_q[\log p(z)] + \\ &\quad (\lambda_1 - \alpha - 1)D_{\text{KL}}(q(z)\|p(z)) + \lambda_2 D_{\text{KL}}(p(z)\|q(z))\end{aligned}$$

*Then any unary likelihood free computable objective of the KL Info-Autoencoding family is equivalent by elementary transformation to a linear combination of $\mathcal{L}_{\text{InfoGAN}}$ and $\mathcal{L}_{\text{InfoVAE}}$.*

Note that there is a slight asymmetry between InfoGAN and InfoVAE because $\log q(x)$ is assumed to be intractable, while $\log p(z)$ is assumed tractable. The consequence is that for InfoGAN, the coefficient for the mutual information term $I_p(x; z)$ must be tied together with the coefficient for $D_{\text{KL}}(q(x|z)\|p(x|z))$ to meet the computational constraint. This means that InfoGAN can only maximize mutual information. On the other hand, InfoVAE can either maximize or minimize mutual information.

Finally for the most general case, we show that any binary likelihood free computable member of the KL Information-Autoencoding family can be written as a linear combination of InfoVAE and an information maximizing/minimizing form of BiGAN/ALI/ALICE (Donahue et al., 2016; Dumoulin et al., 2016b; Li et al., 2017a), which we term InfoBiGAN.

**Theorem 3.** *Define the following objective*

*5) KL InfoBiGAN, where $\alpha, \lambda_1, \lambda_2$ are any real number*

$$\begin{aligned}\mathcal{L}_{\text{InfoBiGAN}} &= \alpha I_p(x; z) + D_{\text{KL}}(p(z|x)\|q(z|x)) + \lambda_1 D_{\text{KL}}(p(x)\|q(x)) + \lambda_2 D_{\text{KL}}(q(x)\|p(x)) \\ &= (\alpha + 1)D_{\text{KL}}(p(x, z)\|q(x, z)) + (\lambda_1 - \alpha - 1)D_{\text{KL}}(p(x)\|q(x)) + \\ &\quad \lambda_2 D_{\text{KL}}(q(x)\|p(x)) - \alpha \mathbb{E}_p[\log q(z|x)]\end{aligned}$$

*Then any binary likelihood free computable objective of the KL Info-Autoencoding family is equivalent by elementary transformation to a linear combination of $\mathcal{L}_{\text{InfoVAE}}$ and $\mathcal{L}_{\text{InfoBiGAN}}$.*

### 3.4 EXTENSIONS AND LIMITATIONS

A simple extension beyond KL divergence is possible. For any KL Info-Autoencoding family under the given tractability assumptions, we may add terms in the same tractability category and the resulting objective will still satisfy the same tractability assumptions. For example, for a subset of the KL InfoVAE family, we can add the Maximum Mean Discrepancy (MMD) distance to derive the MMD InfoVAE (Zhao et al., 2017) in Lagrangian form.

$$\mathcal{L}_{\text{MMD}-\text{VAE}} = \alpha I_q(x; z) + D_{\text{KL}}(q(x|z)\|p(x|z)) + \lambda D_{\text{MMD}}(p(z)\|q(z))$$

and this will still be unary likelihood free computable.

There are two limitations to our analysis. First we do not prove uniqueness results for families with arbitrary divergences. Second we use restricted computability toolboxes and elementary transformations. More general results could be derived by considering a broader family of divergences, computable terms, and transformations. For example, enumeration for models including Renyi divergence variational inference (Li and Turner, 2016) and Stein variational gradient (Liu and Wang, 2016) are interesting directions for future work.

## 4 APPLICATIONS

The theory developed in the previous section reveals the tradeoffs between modeling flexibility and computational costs. We may gain additional insight into the behavior of the models by analyzing another trade-off: the Lagrangian dual parameters $\boldsymbol{\lambda}$ and the information preference parameters $\alpha_1, \alpha_2$ specify how different terms are weighted or "priced" when it is not possible to optimally minimize all of them simultaneously. This gives us insight into the behavior of the models under finite capacity conditions.

### 4.1 CORRECTNESS OF LAGRANGIAN DUAL

In theory, we *should* optimize $\mathcal{L}$ over both $\theta, \phi$ (minimize) *and* $\boldsymbol{\lambda}$ (maximize) to solve the original primal optimization problem (assuming strong duality holds, or to get the tightest relaxation). However, as we have shown in Section 3, that this is not always possible, as only some choices of $\boldsymbol{\lambda}$ lead to a dual with a tractable equivalent form (so that some terms cancel out). In addition, many existing models optimize the Lagrangian dual over $\theta, \phi$ only for a single choice of $\boldsymbol{\lambda}$ but still achieve impressive log likelihoods or sample quality, indicating that the primal is "approximately" solved.

Intuitively when $f(\theta, \phi)$ is lower bounded, and when the model families $\{p_\theta\}$ and $\{q_\phi\}$ are highly flexible, we may achieve both near-optimality for the primal objective $f(\theta, \phi)$, and near-satisfaction of the feasibility constraints. Therefore it might be acceptable to use a single value of $\boldsymbol{\lambda}$, as long as $\boldsymbol{\lambda} > 0$. On the other hand, if $f(\theta, \phi)$ is not lower bounded, regardless of model capacity, for any fixed Lagrangian parameter $\boldsymbol{\lambda}$, the Lagrangian can be minimized over $\theta, \phi$ by only driving down $f(\theta, \phi)$ without lowering $D(p(x, z)||q(x, z))$. It is even possible that $D(p(x, z)||q(x, z))$ may increase if minimizing $f(\theta, \phi)$ is in conflict with minimizing $D(p(x, z)||q(x, z))$.

We verify this intuition experimentally for a simple scenario of a mixture of two Gaussians, where it is clear that a reasonably large neural network is flexible enough to fit the distributions.[6]

$$q(x) \sim 1/2\mathcal{N}(-1, 1/4) + 1/2\mathcal{N}(1, 1/4)$$

We use the $\beta$-VAE model class (Higgins et al., 2016). As discussed in Section 2 and Section 3, this is a Lagrangian dual to the following optimization problem

$$\begin{aligned} \min \quad & (\beta - 1)I_q(x; z) \\ s.t. \quad & D_{\text{KL}}(q(z)\|p(z)) = 0, \qquad D_{\text{KL}}(q(x|z)\|p(x|z)) = 0 \end{aligned}$$

When $\beta < 1$ the optimization objective $(\beta - 1)I_q(x; z)$ is unbounded below for continuous distributions. So optimizing the Lagrangian dual with fixed Lagrangian multipliers may be problematic.

Figure 1 shows the mutual information and negative log likelihood during training. It can be observed that when $\beta < 1$ mutual information grows quickly initially (left panel), while at the same

---

[6]Experimental setting provided in Appendix.

time the log- likelihood (right panel) actually gets worse. However, as soon as mutual information stops growing, either due to finite model capacity (solid line), or when we explicitly upper bound it (dashed line), log-likelihood of the data starts to improve (around iteration 20000 for dashed lines, and around iteration 40000 for solid lines). In the latter case, true log-likelihood actually becomes as good as the one for $\beta = 1$, which is the original ELBO. However, choosing $\beta < 1$ results in significantly larger mutual information $I_q(x; z)$, which might be desirable (Chen et al., 2016b; Zhao et al., 2017).

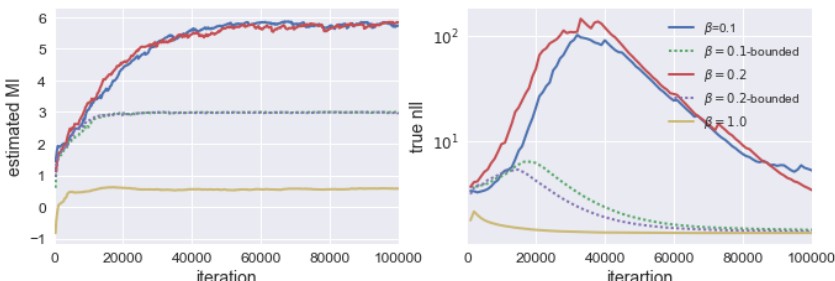

Figure 1: Estimated mutual information (Left) vs. true negative log likelihood (Right). Dashed lines correspond to the same objective but with an upper bound on mutual information term. When mutual information increases, log likelihood gets worse. As soon as mutual information stops increasing, log likelihood starts to improve.

This example illustrates what we believe is a general pattern. With highly flexible $\{p_\theta\}$ and $\{q_\phi\}$, a fixed (suboptimal) $\lambda$ can generally be used if the primal objective has an achievable lower bound. As soon as the model approaches this lower bound, it will focus on satisfying the consistency conditions.

## 4.2 TRADE-OFF FOR INFEASIBLE OPTIMIZATION PROBLEMS

All objectives of the Info-Autoencoding family are Lagrangian duals of the same primal problem. However, different choices for $\lambda$ may lead to very different behaviors when optimality cannot be achieved, either due to limited model capacity or optimization difficulties. The Lagrangian parameters $\lambda$ give the cost of violating each constraint, so that when not all constraints can be satisfied, the model will prioritize constraints with larger Lagrangian coefficients. We use this perspective to analyze the benefit of the additional flexibility gained by giving up likelihood based computability.

For example, we consider the consistency condition $D(q(z)\|p(z)) = 0, D(q(x|z)\|p(x|z)) = 0$. There is an asymmetry between the input space $\mathcal{X}$ and the latent feature space $\mathcal{Z}$. $\mathcal{X}$ is usually high dimensional, while $\mathcal{Z}$ is low dimensional. Therefore any mistake fitting $\mathcal{X}$ is likely to be magnified. For example, consider fitting an $n$ dimensional distribution $\mathcal{N}(0, I)$ with $\mathcal{N}(\epsilon, I)$ using KL divergence:

$$D_{\mathrm{KL}}(\mathcal{N}(0, I), \mathcal{N}(\epsilon, I)) = n\epsilon^2/2$$

As $n$ increases with some fixed $\epsilon$, the Euclidean distance between the mean of the two distributions is $\Theta(\sqrt{n})$, yet the corresponding $D_{\mathrm{KL}}$ becomes $\Theta(n)$. For natural images, the dimensionality of $\mathcal{X}$ is often orders of magnitude larger than the dimensionality of $\mathcal{Z}$. This means that the model will tend to sacrifice consistency between $p(z)$ and $q(z)$ in order to fit $q(x|z)$ and $p(x|z)$. We may wish to calibrate this with a larger multiplier for the $D(q(z)\|p(z)) = 0$ constraints. We show that doing so improves performance metrics such as log likelihood, but it is only feasible with likelihood free optimization.

We consider the following two classes of models written in Lagrangian dual form:

$$\mathcal{L}_{\beta-\mathrm{ELBO}} = (\beta - 1)I_q(x; z) + \beta D_{\mathrm{KL}}(q(z)\|p(z)) + D_{\mathrm{KL}}(q(x|z)\|p(x|z))$$
$$\mathcal{L}_{\mathrm{InfoVAE}} = \mathcal{L}_{\beta-\mathrm{ELBO}} + \lambda D(q(z)\|p(z))$$

We use a slightly different but equivalent definition from Theorem 2 to highlight the fact that InfoVAE has an additional free parameter $\lambda$. Note that the additional degree of freedom is not possible for $\beta$-ELBO as a result of the likelihood based requirement; if we increase the weighting for

$D_{\mathrm{KL}}(q(z)\|p(z))$ we must penalize the mutual information term by the same amount to maintain likelihood-based tractability.

We compute an estimate of the true log likelihood over the binarized MNIST dataset, where the objectives are $\mathcal{L}_{\beta-\mathrm{ELBO}}$ and $\mathcal{L}_{\mathrm{InfoVAE}}$ respectively. Here $\mathcal{L}_{\beta-\mathrm{ELBO}}$ only depends on $\beta$, whereas $\mathcal{L}_{\mathrm{InfoVAE}}$ is allowed to select the $\lambda$ parameter, allowing for an additional degree of freedom [7]. We can choose $\lambda > 0$ which encourages $q(z)$ and $p(z)$ to be close. Interestingly, even for a very large $\lambda = 5000$ the InfoVAE objective is still able to perform reasonably well, which is due to the fact that $z$ has a much smaller dimension than $x$ (50 compared to 784 dimensions). As shown in Figure 2, the additional degree of freedom provided by $\lambda$ is able to result in better density estimation in both training and test sets, even when $\beta$ takes suboptimal values indicating either too much or too little mutual information between $x$ and $z$. This is surprising given that for $\lambda > 0$ or $\beta < 1$, we are no longer optimizing over a lower bound on the log-likelihood directly.

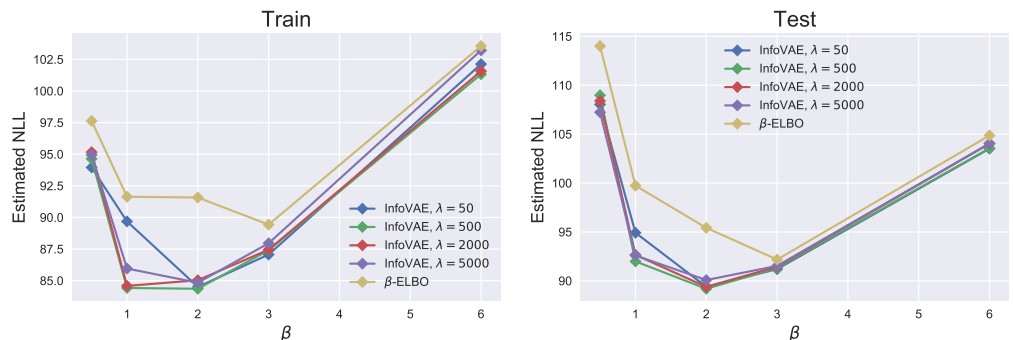

Figure 2: Estimated negative log likelihood under different $\beta$ and $\lambda$ values (lower is better). For both training set (left) and test set (right) the extra degree of freedom from $\lambda$ improves performance (the curresponding curves are below the yellow one, for all values of $\beta$).

## 5    CONCLUSION

In this paper we explored a large family of learning objectives and models that generalize ELBO, $\beta$-ELBO, BiGAN/ALI/ALICE, InfoGAN, AVB, CycleGAN/DiscoGAN, AAE/InfoVAE. The Lagrangian form reveals several important properties of the model, while the corresponding "tractable" form tells us how we may optimize it. Under some assumptions, we were able to identify the most general class of objectives that can be optimized using likelihood-based and likelihood-free methods.

Future work includes exploring additional model classes, generalizing the analysis beyond two groups of variables, and extending the tractability analysis beyond KL divergence and elementary transformations.

Since the Information-Autoencoding Family introduces a wide range of valid objective configurations through $\alpha_1, \alpha_2, \boldsymbol{\lambda}$ parameters, it is also interesting to see scalable methods to optimize/search for the most effective set of parameters. Bayesian optimization methods Snoek et al. (2012) and recent advances in neural architecture search (Zoph and Le, 2016; Brock et al., 2017) might be useful for this purpose.

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

## A    EXAMPLES OF UNDER-DETERMINATION OF CONSISTENCY CONDITION

**Example 1**: Let $\mathcal{X}$ and $\mathcal{Z}$ be discrete sets with $n$ distinct elements. Then $p_\theta(x|z), q_\phi(z|x)$ are tables with $n(n-1)$ degrees of freedom each, giving a total of $2n(n-1)$ free parameters. The requirement $p_\theta(x) = p_{\text{data}}(x)$ for all $x \in \mathcal{X}$ involves $n-1$ linear constraints, while $q_\phi(z|x) = p_\theta(z|x)$ for all $x$ adds additional $n(n-1)$ linear constraints. Therefore, there are at least

$$2n(n-1) - (n+1)(n-1) = (n-1)^2$$

free parameters satisfying consistency constraints. Therefore for all $n \geq 2$, there are multiple consistent solutions.

**Example 2**: If for any $z$, $p_\theta(x|z)$ is itself a sufficiently flexible distribution family (e.g. represented by PixelCNN), then multiple solutions may exists. For example, one solution could be

$$p_\theta(x|z) = q(x), \ \forall z \in \text{support}(q(x)) \quad q_\phi(z|x) = p(z), \ \forall x \in \text{support}(p(z))$$

which means that input $x$ and latent code $z$ are completely independent. This may be an undesirable solution, and several models have been proposed to avoid this (Chen et al., 2016b; Zhao et al., 2017).

## B    PROOFS

*Proof of Corollary 1.* Assume they are consistent. Then for any $x$ such that $p_{data}(x) \neq 0$ and for any $z$,

$$q_\phi(x,z) = p_{\text{data}}(x)q_\phi(z|x) = p_\theta(x)q_\phi(z|x) = p_\theta(x)p_\theta(z|x)$$

For any $x$ such that $p_{data}(x) = 0$ and for any $z$,

$$q_\phi(x,z) = p_{\text{data}}(x)q_\phi(z|x) = 0$$

and because $0 = p_{\text{data}}(x) = p_\theta(x) = \sum_z p_\theta(x,z)$, it follows that $p(x,z) = 0$ for all $z$.

Assuming $q_\phi(x,z) = p_\theta(x,z)$ we have that

$$p_\theta(x) = \sum_z p_\theta(x,z) = \sum_z q_\phi(x,z) = p_{\text{data}}(x)$$

Thus, the conditionals need to match when they are defined, i.e., when $p_{\text{data}}(x) = p_\theta(x) \neq 0$.    □

*Proof of Theorem 1 2 3.* If all divergences are KL or reverse KL, then we can enumerate over every possible mutual information or divergence term in Definition 5.

$$
\begin{array}{cc}
I_p(x;z) & I_q(x;z) \\
D_{\text{KL}}(p(x,z)||q(x,z)) & D_{\text{KL}}(q(x,z)||p(x,z)) \\
D_{\text{KL}}(p(x)||q(x)) & D_{\text{KL}}(q(x)||p(x)) \\
D_{\text{KL}}(p(z)||q(z)) & D_{\text{KL}}(q(z)||p(z)) \\
\mathbb{E}_p[D_{\text{KL}}(p(z|x)||q(z|x))] & \mathbb{E}_q[D_{\text{KL}}(q(z|x)||p(z|x))] \\
\mathbb{E}_p[D_{\text{KL}}(p(x|z)||q(x|z))] & \mathbb{E}_q[D_{\text{KL}}(q(x|z)||p(x|z))]
\end{array}
\tag{16}
$$

Note that we do not consider the terms

$$
\begin{array}{cc}
\mathbb{E}_q[D_{\text{KL}}(p(z|x)||q(z|x))] & \mathbb{E}_p[D_{\text{KL}}(q(z|x)||p(z|x))] \\
\mathbb{E}_q[D_{\text{KL}}(p(x|z)||q(x|z))] & \mathbb{E}_p[D_{\text{KL}}(q(x|z)||p(x|z))]
\end{array}
\tag{17}
$$

because these terms are not in any tractability toolbox in Definition 7, nor can any elementary transformation be applied to them. So any KL Info-Autoencoding objective that contain any of these terms do not have a equivalent computable form. We only consider the 12 possibilities in (16).

To prove this theorem we first map all optimization objectives into a vector space $V \subseteq (\Theta \times \Phi \to \mathbb{R})$ with standard function addition and scalar multiplication as operations. Here $\Theta$ is the domain for $\theta$, $\Phi$ is the domain for $\phi$, and $f : \Theta \times \Phi \to \mathbb{R}$ represents the objective function.

Define a set of basic terms

$$\boldsymbol{p}_{\text{atoms}} = \begin{pmatrix} \log p(x,z) \\ \log p(x|z) \\ \log p(z|x) \\ \log p(x) \\ \log p(z) \end{pmatrix} \qquad \boldsymbol{q}_{\text{atoms}} = \begin{pmatrix} \log q(x,z) \\ \log q(x|z) \\ \log q(z|x) \\ \log q(x) \\ \log q(z) \end{pmatrix}$$

Define the following set of basis vectors with 20 objective functions

$$\boldsymbol{E} = \begin{pmatrix} \mathbb{E}_{p(x,z)}[\boldsymbol{p}_{\text{atoms}}] \\ \mathbb{E}_{p(x,z)}[\boldsymbol{q}_{\text{atoms}}] \\ \mathbb{E}_{q(x,z)}[\boldsymbol{p}_{\text{atoms}}] \\ \mathbb{E}_{q(x,z)}[\boldsymbol{q}_{\text{atoms}}] \end{pmatrix}^T$$

It is easy to verify that all objectives of the KL Information Autoencoding family in (16), tractable terms in Definition 7 and transformations defined Definition 8 can be written as linear combinations of this set of basis, so they belong to the vector space spanned by these basis vectors. Therefore any optimization objective $\mathcal{L}$ in our family can be represented as a vector $\boldsymbol{n} \in \mathbb{R}^{20}$, where

$$\mathcal{L} = \boldsymbol{En}$$

For any $m \in \mathbb{N}$, the column space of any matrix $N \in \mathbb{R}^{n \times m}$ can be used to represent a family of objectives. Each column $\boldsymbol{n}_i$ of $N$ corresponds to an objective $\boldsymbol{En}_i$, and the family is defined by any linear combination of $\{\boldsymbol{En}_1, \cdots, \boldsymbol{En}_m\}$ Therefore we have embedded optimization objectives and linear model families into vectors and column spaces of matrices in this vector space. Now we write each objective and transformation we have discussed as vectors and matrices in this space.

**1) KL Info-Autoencoding family**.

Let $R \in \mathbb{R}^{20 \times 12}$ be

$$\begin{pmatrix}
0 & 1 & 0 & 0 & 0 & 0 & & & & & & \\
0 & 0 & 1 & 0 & 0 & 0 & & & & & & \\
1 & 0 & 0 & 1 & 0 & 0 & & & & & & \\
0 & 0 & 0 & 0 & 1 & 0 & & & & & & \\
-1 & 0 & 0 & 0 & 0 & 1 & & & & & & \\
0 & -1 & 0 & 0 & 0 & 0 & & & \mathbf{0} & & & \\
0 & 0 & -1 & 0 & 0 & 0 & & & & & & \\
0 & 0 & 0 & -1 & 0 & 0 & & & & & & \\
0 & 0 & 0 & 0 & -1 & 0 & & & & & & \\
0 & 0 & 0 & 0 & 0 & -1 & & & & & & \\
& & & & & & 0 & 1 & 0 & 0 & 0 & 0 \\
& & & & & & 0 & 0 & 1 & 0 & 0 & 0 \\
& & & & & & 0 & 0 & 0 & 1 & 0 & 0 \\
& & & & & & 0 & 0 & 0 & 0 & 1 & 0 \\
& & & \mathbf{0} & & & 0 & 0 & 0 & 0 & 0 & 1 \\
& & & & & & 0 & -1 & 0 & 0 & 0 & 0 \\
& & & & & & 0 & 0 & -1 & 0 & 0 & 0 \\
& & & & & & 1 & 0 & 0 & -1 & 0 & 0 \\
& & & & & & 0 & 0 & 0 & 0 & -1 & 0 \\
& & & & & & -1 & 0 & 0 & 0 & 0 & -1
\end{pmatrix}$$

The column space of this matrix form a basis where each column is one of the twelve terms defined in Eq.(16). Denote the subspace spanned by the columns as $\mathcal{R}$. This subspace represents all objectives that are a linear combination of Eq.(16), or equivalently, every possible objective in Lagrangian dual form in Definition 5 where each divergence is either KL or reverse KL and does not include terms in (17).

**2) Equivalent transformations**. Note that all elementary transformations are log linear. For example, replacing $\mathbb{E}_p[\log p(x,z)]$ with $\mathbb{E}_p[\log p(x|z)] + \mathbb{E}_p[\log p(z)]$ leaves any objective equivalent. This means that for any $b \in \mathbb{R}$

$$\boldsymbol{n}' = \boldsymbol{n} + b(1, -1, 0, 0, -1, 0, \cdots, 0)^T$$

Results in

$$\boldsymbol{En} \equiv \boldsymbol{En}'$$

Enumerating over all 10 equivalent transformations we considered in Definition 8, we define the matrix $P \in \mathbb{R}^{20 \times 10}$ as

$$
\begin{pmatrix}
1 & 1 & 0 & & & & & & \\
-1 & 0 & 0 & & & & & & \\
0 & -1 & 0 & & & & & & \\
0 & -1 & 0 & & & & & & \\
-1 & 0 & 1 & & & & 0 & & \\
& & & 1 & 1 & & & & \\
& & & -1 & 0 & & & & \\
& & & 0 & -1 & & & & \\
& & & 0 & -1 & & & & \\
& & & -1 & 0 & & & & \\
& & & & & 1 & 1 & & \\
& & & & & -1 & 0 & & \\
& & & & & 0 & -1 & & \\
& & & & & 0 & -1 & & \\
& & & & & -1 & 0 & & \\
& 0 & & & & & & 1 & 1 & 0 \\
& & & & & & & -1 & 0 & 0 \\
& & & & & & & 0 & -1 & 0 \\
& & & & & & & 0 & -1 & 1 \\
& & & & & & & -1 & 0 & 0
\end{pmatrix}
$$

It is easy to see that for each column $\boldsymbol{c}$ of P, $\boldsymbol{n} + \boldsymbol{c}$ corresponds to performing one of the elementary transformations in Definition 8. Any sequence of elementary transformations can be written as $\boldsymbol{n} + P\boldsymbol{b}$ for some vector $\boldsymbol{b}$.

**3) Tractable Families**. Tractable families are given by Definition 7. We find the corresponding basis matrix $T$ and subspace $\mathcal{T}$ for each tractability assumption in the definition. Then any tractable objective under elementary transformation lie in the subspace $\mathcal{T} + \mathcal{P}$. $+$ is the sum of vector spaces $\mathcal{T}$ and $\mathcal{P}$, or space of all vectors that can be written as the linear combination of vectors in $\mathcal{T}$ and vectors in $\mathcal{P}$.

For likelihood based computable objectives, if is easy to see that the eight likelihood based computable terms in Definition 7 can be written as the columns of $T_{\mathrm{lb}}$ defined as

$$
T_{\mathrm{lb}}^{p} = T_{\mathrm{lb}}^{q} =
\begin{pmatrix}
1 & 0 & 0 & 0 \\
0 & 1 & 0 & 0 \\
0 & 0 & 0 & 0 \\
0 & 0 & 0 & 0 \\
0 & 0 & 1 & 0 \\
0 & 0 & 0 & 0 \\
0 & 0 & 0 & 0 \\
0 & 0 & 0 & 1 \\
0 & 0 & 0 & 0 \\
0 & 0 & 0 & 0
\end{pmatrix}
\qquad
T_{\mathrm{lb}} =
\begin{pmatrix}
T_{\mathrm{lb}}^{p} & 0 \\
0 & T_{\mathrm{lb}}^{q}
\end{pmatrix}
$$

denote the corresponding subspace as $\mathcal{T}_{\mathrm{lb}}$.

For unary likelihood free computable objectives, aside from all likelihood based computable objectives, we also add the four divergences

$$D_{\mathrm{KL}}(p(x)\|q(x)), D_{\mathrm{KL}}(q(x)\|p(x)), D_{\mathrm{KL}}(p(z)\|q(z)), D_{\mathrm{KL}}(q(z)\|p(z))$$

similarly we have

$$T_{\text{ulf}}^p = T_{\text{ulf}}^q = \begin{pmatrix} & 0 & 0 \\ & 0 & 0 \\ & 0 & 0 \\ & -1 & 0 \\ T_{\text{lb}} & 0 & -1 \\ & 0 & 0 \\ & 0 & 0 \\ & 0 & 0 \\ & 1 & 0 \\ & 0 & 1 \end{pmatrix} \quad T_{\text{ulf}} = \begin{pmatrix} T_{\text{ulf}}^p & 0 \\ 0 & T_{\text{ulf}}^q \end{pmatrix}$$

All unary likelihood free computable objectives are columns of $T_{\text{ulf}}$, denote the corresponding subspace as $\mathcal{T}_{\text{ulf}}$.

For binary likelihood free computable objectives, aside from all unary likelihood free computable objectives, we also add the two divergences

$$D_{\text{KL}}(p(x,z)||q(x,z)), D_{\text{KL}}(q(x,z)||p(x,z))$$

similarly we have

$$T_{\text{blf}}^p = T_{\text{blf}}^q = \begin{pmatrix} & 1 \\ & 0 \\ & 0 \\ & 0 \\ T_{\text{ulf}} & 0 \\ & -1 \\ & 0 \\ & 0 \\ & 0 \\ & 0 \end{pmatrix} \quad T_{\text{blf}} = \begin{pmatrix} T_{\text{blf}}^p & 0 \\ 0 & T_{\text{blf}}^q \end{pmatrix}$$

All binary likelihood free computable objectives are columns of $T_{\text{blf}}$, denote the corresponding subspace as $\mathcal{T}_{\text{blf}}$.

**4) Known Objective Families**. We write out the subspace spanned by the two likelihood based computable family $\beta$-ELBO and Variational mutual information maximization

$$\begin{pmatrix} S_{\text{VMI}} & S_{\beta-\text{ELBO}} \end{pmatrix} = \begin{pmatrix} 0 & \\ 0 & \\ 0 & \\ 0 & \\ 1 & \\ 0 & 0 \\ 0 & \\ -1 & \\ 0 & \\ 0 & \\ & 0 & 0 \\ & -1 & 0 \\ & 0 & 0 \\ & 0 & 0 \\ 0 & 0 & -1 \\ & 0 & 0 \\ & 0 & -1 \\ & 0 & 0 \\ & 1 & 1 \\ & 0 & 1 \end{pmatrix}$$

Denote the corresponding column space as $\mathcal{S}_{\text{VMI}} + \mathcal{S}_{\beta-\text{ELBO}}$.

For subspace spanned by InfoGAN and InfoVAE we have ( $S_{\text{InfoGAN}}$   $S_{\text{InfoVAE}}$ ) as

$$
\begin{pmatrix}
0 & 0 & & 0 & & & \\
-1 & 0 & & 0 & & & \\
1 & 0 & & 0 & & & \\
1 & 1 & & -1 & & & \\
0 & 0 & \ 0\  & 0 & & 0 & \\
0 & 0 & & 0 & & & \\
0 & 0 & & 0 & & & \\
-1 & 0 & & 0 & & & \\
0 & -1 & & 1 & & & \\
0 & 0 & & 0 & & & \\
 & & & 0 & & 0 & 0 & 0 \\
 & & & 0 & & 0 & -1 & 0 \\
 & & & 0 & & 0 & 0 & 0 \\
 & & & 0 & & 0 & 0 & 0 \\
 & 0 & & 1 & \ 0\  & 0 & 0 & -1 \\
 & & & 0 & & 0 & 0 & 0 \\
 & & & 0 & & 0 & 1 & 0 \\
 & & & 0 & & 1 & 0 & 0 \\
 & & & 0 & & 0 & 0 & 0 \\
 & & & -1 & & -1 & 0 & 1 \\
\end{pmatrix}
$$

Denote the corresponding column space as $S_{\text{InfoGAN}} + S_{\text{InfoVAE}}$.

For subspace spanned by InfoBiGAN and InfoVAE we have ( $S_{\text{InfoBiGAN}}$   $S_{\text{InfoVAE}}$ ) as

$$
\begin{pmatrix}
0 & 0 & 0 & & 0 & & & \\
1 & 0 & 0 & & 0 & & & \\
0 & 1 & 0 & & 0 & & & \\
-1 & 0 & 1 & & -1 & & & \\
0 & 0 & 0 & \ 0\  & 0 & & 0 & \\
0 & 0 & 0 & & 0 & & & \\
0 & 0 & 0 & & 0 & & & \\
0 & -1 & 0 & & 0 & & & \\
0 & 0 & -1 & & 1 & & & \\
0 & 0 & 0 & & 0 & & & \\
 & & & & 0 & & 0 & 0 & 0 \\
 & & & & 0 & & 0 & -1 & 0 \\
 & & & & 0 & & 0 & 0 & 0 \\
 & & & & 0 & & 0 & 0 & 0 \\
 & 0 & & & 1 & \ 0\  & 0 & 0 & -1 \\
 & & & & 0 & & 0 & 0 & 0 \\
 & & & & 0 & & 0 & 1 & 0 \\
 & & & & 0 & & 1 & 0 & 0 \\
 & & & & 0 & & 0 & 0 & 0 \\
 & & & & -1 & & -1 & 0 & 1 \\
\end{pmatrix}
$$

Denote the corresponding column space as $S_{\text{InfoBiGAN}} + S_{\text{InfoVAE}}$.

**5) Subspace Equivalence Relationship**.

As before denote $\mathcal{A} + \mathcal{B}$ as the sum of two subspaces $\mathcal{A}$ and $\mathcal{B}$, or all vectors that are linear combinations of vectors in $\mathcal{A}$ and vectors in $\mathcal{B}$. Consider first the $\mathcal{L}_{\beta-\text{ELBO}}$ family and $\mathcal{L}_{VMI}$ family.

$$
\mathcal{L}_{\text{VMI}} = -\lambda_1 I_p(x; z) + \lambda_1 D_{\text{KL}}(p(z|x)\|q(z|x)) \tag{18}
$$

$$
= \lambda_1 \mathbb{E}_p[\log q(z|x)] \tag{19}
$$

$$
\mathcal{L}_{\beta-\text{ELBO}} = (\lambda_2 - \lambda_1) I_q(x, z) + \lambda_1 D_{\text{KL}}(q(x|z)\|p(x|z)) + \lambda_2 D_{\text{KL}}(q(z)\|p(z)) \tag{20}
$$

$$
= -\lambda_1 \mathbb{E}_q[\log p(x|z)] + \lambda_2 D_{\text{KL}}(q(z|x)\|p(z)) \tag{21}
$$

Notice that any member of the $\mathcal{L}_{\beta-\text{ELBO}}$ family and $\mathcal{L}_{VMI}$ family can be transformed by elementary transformation to a Lagrangian dual form Eq.(18)(20), or to a tractable form that only contains

terms in $\mathcal{T}_{lb}$ in Eq.(19)21). This means that for any such member $s$, there is a $r$ in the space of all Lagrangian dual forms $\mathcal{R}$, and a $p$ in space of all transformations $\mathcal{P}$ so that $s = r + p$. Therefore

$$\mathcal{S}_{\beta-\mathrm{ELBO}} + \mathcal{S}_{VMI} \subset \mathcal{R} + \mathcal{P}$$

Similarly, for any such member $s$, there is a $t \in \mathcal{T}_{\mathrm{lb}}$, and $p \in \mathcal{P}$ so that $s = t + p$. so

$$\mathcal{S}_{\beta-\mathrm{ELBO}} + \mathcal{S}_{VMI} \subset \mathcal{T}_{\mathrm{lb}} + \mathcal{P}$$

Therefore

$$\mathcal{S}_{\beta-\mathrm{ELBO}} + \mathcal{S}_{VMI} \subset (\mathcal{R} + \mathcal{P}) \cap (\mathcal{T}_{\mathrm{lb}} + \mathcal{P})$$

In addition we have

$$\mathcal{P} \subset (\mathcal{R} + \mathcal{P}) \cap (\mathcal{T}_{\mathrm{lb}} + \mathcal{P})$$

so

$$\mathcal{S}_{\beta-\mathrm{ELBO}} + \mathcal{S}_{\mathrm{VMI}} + \mathcal{P} \subset (\mathcal{R} + \mathcal{P}) \cap (\mathcal{T}_{\mathrm{lb}} + \mathcal{P}) \tag{22}$$

Therefore if we can further have

$$\dim(\mathcal{S}_{\beta-\mathrm{ELBO}} + \mathcal{S}_{\mathrm{VMI}} + \mathcal{P}) = \dim\left((\mathcal{R} + \mathcal{P}) \cap (\mathcal{T}_{\mathrm{lb}} + \mathcal{P})\right)$$

the above subspaces must be equivalent. This is because if the two spaces are not equivalent, and there exists vector $\boldsymbol{v} \in (\mathcal{R} + \mathcal{P}) \cap (\mathcal{T}_{\mathrm{lb}} + \mathcal{P})$ but $\boldsymbol{v} \notin \mathcal{S}_{\beta-\mathrm{ELBO}} + \mathcal{S}_{\mathrm{VMI}} + \mathcal{P}$. However

$$\mathcal{S}_{\beta-\mathrm{ELBO}} + \mathcal{S}_{\mathrm{VMI}} + \mathcal{P} + \{\boldsymbol{v}\} \subset (\mathcal{R} + \mathcal{P}) \cap (\mathcal{T}_{\mathrm{lb}} + \mathcal{P})$$

But

$$\begin{aligned} \dim\left(\mathcal{S}_{\beta-\mathrm{ELBO}} + \mathcal{S}_{\mathrm{VMI}} + \mathcal{P} + \{\boldsymbol{v}\}\right) > \\ \dim\left(\mathcal{S}_{\beta-\mathrm{ELBO}} + \mathcal{S}_{\mathrm{VMI}} + \mathcal{P}\right) = \dim\left((\mathcal{R} + \mathcal{P}) \cap (\mathcal{T}_{\mathrm{lb}} + \mathcal{P})\right) \end{aligned}$$

leading to a contradiction.

Similarly we have for the other families

$$\mathcal{S}_{\mathrm{InfoGAN}} + \mathcal{S}_{\mathrm{InfoVAE}} + \mathcal{P} \subset (\mathcal{R} + \mathcal{P}) \cap (\mathcal{T}_{\mathrm{ulf}} + \mathcal{P}) \tag{23}$$
$$\mathcal{S}_{\mathrm{InfoBiGAN}} + \mathcal{S}_{\mathrm{InfoVAE}} + \mathcal{P} \subset (\mathcal{R} + \mathcal{P}) \cap (\mathcal{T}_{blf} + \mathcal{P}) \tag{24}$$

To compute the above dimensions, we compute a basis for the each space under discussion. The rank of basis matrix is equal to the dimensionality of its column space. Let $\mathcal{A}, \mathcal{B}$ be the column space of matrices $A, B$, then the basis matrix of $\mathcal{A} + \mathcal{B}$ can be derived by $(\begin{array}{cc} A & B \end{array})$. the basis matrix of $\mathcal{A} \cap \mathcal{B}$ can be derived by the following process: Compute the null space for

$$\left(\begin{array}{cc} A & B \end{array}\right)\left(\begin{array}{c} \boldsymbol{u} \\ \boldsymbol{v} \end{array}\right) = 0$$

Let $U, V$ be basis matrix for the null space of $\boldsymbol{u}, \boldsymbol{v}$ respectively. Then $AU = -BV$ is a basis matrix for $\mathcal{A} \cap \mathcal{B}$.

We can perform this procedure for all the above mentioned subspaces, and derive the following table.

| subspace | dimension |
|---|---|
| $(\mathcal{S}_{\beta-\mathrm{ELBO}} + \mathcal{S}_{\mathrm{VMI}}) + \mathcal{P}$ | 13 |
| $(\mathcal{T}_{lb} + \mathcal{P}) \cap (\mathcal{R} + \mathcal{P})$ | 13 |
| $(\mathcal{S}_{\mathrm{InfoGAN}} \cup \mathcal{S}_{\mathrm{InfoVAE}}) + \mathcal{P}$ | 17 |
| $(\mathcal{T}_{ulf} + \mathcal{P}) \cap (\mathcal{R} + \mathcal{P})$ | 17 |
| $(\mathcal{S}_{\mathrm{InfoBiGAN}} \cup \mathcal{S}_{\mathrm{InfoVAE}}) + \mathcal{P}$ | 18 |
| $(\mathcal{T}_{blf} + \mathcal{P}) \cap (\mathcal{R} + \mathcal{P})$ | 18 |

Therefore we have verified that dimensionality match for Eq. (22, 23, 24). Therefore we have

$$\mathcal{S}_{\beta-\text{ELBO}} + \mathcal{S}_{\text{VMI}} + \mathcal{P} = (\mathcal{R} + \mathcal{P}) \cap (\mathcal{T}_{\text{lb}} + \mathcal{P})$$
$$\mathcal{S}_{\text{InfoGAN}} + \mathcal{S}_{\text{InfoVAE}} + \mathcal{P} = (\mathcal{R} + \mathcal{P}) \cap (\mathcal{T}_{\text{ulf}} + \mathcal{P})$$
$$\mathcal{S}_{\text{InfoBiGAN}} + \mathcal{S}_{\text{InfoVAE}} + \mathcal{P} = (\mathcal{R} + \mathcal{P}) \cap (\mathcal{T}_{blf} + \mathcal{P})$$

This implies that for any objective $r \in \mathcal{R}$ of the Info-Autoencoding family, if $r$ can be converted into likelihood based tractable form by elementary transformations, or $r \in \mathcal{T}_{\text{lb}} + \mathcal{P}$, then

$$r \in \mathcal{R} \cap (\mathcal{T}_{\text{lb}} + \mathcal{P}) \subset (\mathcal{R} + \mathcal{P}) \cap (\mathcal{T}_{\text{lb}} + \mathcal{P})$$

This means

$$r \in \mathcal{S}_{\beta-\text{ELBO}} + \mathcal{S}_{\text{VMI}} + \mathcal{P}$$

which implies that $r$ can be converted by elementary transformation into a linear combination of elements in $\mathcal{S}_{\beta-\text{ELBO}}$ and elements in $\mathcal{S}_{VMI}$. We can derive identical conclusion for the other two objective families. $\square$

## C  EXPERIMENTAL SETUP

### C.1  EXPERIMENTAL SETUP FOR SECTION 4.1

The latent space $z$ is 1-dimensional, and both $p(x|z)$ and $q(z|x)$ are 1-dimensional Gaussians with arbitrary mean and variance. $p(z)$ is the standard Gaussian, and the mapping $z \to p(x|z)$ and $x \to q(z|x)$ are both fully connected neural networks with two hidden layers each with 1024 units.

To estimate mutual information we can use $-\mathbb{E}_q[\log p(x|z)]$ because

$$-\mathbb{E}_q[\log p(x|z)] = I_q(x; z) - D_{\text{KL}}(q(x|z)\|p(x|z))$$

so we get a upper bound on the mutual information $I_q(x; z)$ that is tight if $p(x|z)$ matches $q(x|z)$ well. We also upper bound $-\mathbb{E}_q[\log p(x|z)]$ to place a explicit upper bound on the mutual information. To compute true log likelihood for this toy problem it is sufficient to estimate the integration by sampling

$$\log p(x) = \log \mathbb{E}_{p(z)}[p(x|z)] \approx \log \frac{1}{N} \sum_{i=1,\cdots,N, z_i \sim p(z)} p(x|z_i)$$

### C.2  EXPERIMENTAL SETUP FOR SECTION 4.2

We consider $p(x|z)$ with one layer of latent code, and its corresponding inference network $q(z|x)$. Both $p$ and $q$ have two hidden layers with 1024 neurons each and tanh activations; the dimension of the latent variable $z$ is 50. We obtain a binarized MNIST dataset with 50000 training examples and 10000 test examples by sampling each dimension from a Bernoulli distribution with probability given by the MNIST dataset. We use importance sampling (instead of AIS (Wu et al., 2016))

$$\log p(x) = \log E_{p(z|x)} \left[ \frac{p(x, z)}{q(z|x)} \right] \approx \log \frac{1}{N} \sum_{i=1,\cdots,N, z_i \sim q(z|x)} \frac{p(x, z_i)}{q(z_i|x)}$$

to evaluate the negative log-likelihood. Hence the results reported are slightly worse than the state of the art. This does not affect our arguments though, since we are comparing the relative performance of $\beta$-ELBO and InfoVAE.

