# OpenReview forum: "The Information-Autoencoding Family: A Lagrangian Perspective on Latent Variable Generative Modeling"
_ICLR.cc/2018/Conference — Reject_

### Official Review · AnonReviewer3 · 2017-11-26
**Contains some interesting results but the presentation is not focused**

**Rating:** 6
**Confidence:** 4

**Review:**

Thank you for the feedback, I have read it.

I do think that developing unifying frameworks is important. But not all unifying perspective is interesting; rather, a good unifying perspective should identify the behaviour of existing algorithms and inspire new algorithms.

In this perspective, the proposed framework might be useful, but as noted in the original review, the presentation is not clear, and it's not convincing to me that the MI framework is indeed useful in the sense I described above.

I think probably the issue is the lack of good evaluation methods for generative models. Test-LL has no causal relationship to the quality of the generated data. So does MI. So I don't think the argument of preferring MI over MLE is convincing.

So in summary, I will still keep my original score. I think the paper will be accepted by other venues if the presentation is improved and the advantage of the MI perspective is more explicitly demonstrated.

==== original review ====

Thank you for an interesting read.

The paper presented a unifying framework for many existing generative modelling techniques, by first considering constrained optimisation problem of mutual information, then addressing the problem using Lagrange multipliers.

I see the technical contribution to be the three theorems, in the sense that it gives a closure of all possible objective functions (if using the KL divergences). This can be useful: I'm tired of reading papers which just add some extra "regularisation terms" and claim they work. I did not check every equation of the proof, but it seems correct to me.

However, an imperfection is, the paper did not provide a convincing explanation on why their view should be preferred compared to the original papers' intuition.  For example in VAE case, why this mutual information view is better than the traditional view of approximate MLE, where q is known to be the approximate posterior? A better explanation on this (and similarly for say infoGAN/infoVAE) will significantly improve the paper.

Continuing on the above point, why in section 4 you turn to discuss relationship between mutual information and test-LL?  How does that relate to the main point you want to present in the paper, which is to prefer MI interpretation if I understand it correctly?

Term usage: we usually *maximize* the ELBO and *minimise* the variational free-energy (VFE).

---

> ### Author Response · Authors · 2017-12-28
> **Clarification on Main Concerns**
>
> We thank the reviewers for their time and valuable feedback.
>
> “However, an imperfection is, the paper did not provide a convincing explanation on why their view should be preferred compared to the original papers' intuition.”  For example in VAE case, why this mutual information view is better than the traditional view of approximate MLE, where q is known to be the approximate posterior? A better explanation on this (and similarly for say infoGAN/infoVAE) will significantly improve the paper. Continuing on the above point, why in section 4 you turn to discuss relationship between mutual information and test-LL?  How does that relate to the main point you want to present in the paper, which is to prefer MI interpretation if I understand it correctly?“
>
> Our view (optimize mutual information under distribution matching constraint) provides several understandings traditional perspectives do not provide. First, several attributes of an objective are revealed by the Lagrangian form: information preference, possible optimization methods (likelihood based or likelihood free), closure (most generic form) of model family, etc. In addition Section 4 proceeds to demonstrate two applications where the Lagrangian perspective reveal problems/features that are difficult to identify from traditional perspectives.
>
>
> 1.Correct optimization of the Lagrangian dual requires maximization over the Lagrangian parameters. However, all existing methods use fixed (arbitrarily chosen) Lagrangian parameters. We show failure cases where this does not correctly optimize the primal problem. For example, when the primal objective is information maximization under constraints of distributional consistency, optimization with fixed Lagrangian parameters can maximize mutual information indefinitely without ever encouraging distributional consistency. As a result, data fit (distributional consistency) may even get worse during training (for example, resulting in lower test log likelihood) as mutual information is maximized. We show that this also happens in practice.
>
> 2.The Lagrangian perspective allows us to explicitly weight (“price”) different (conflicting) terms in the objective. For example, suppose the input x has more dimensions than the feature space z. Then for the same per-dimension loss, the input space is weighted more than the latent space (because it has more dimensions). We show in the paper that increasing the weight on matching marginals on z can solve the problem and leads to better performance. In general, we can write out the desired preference in Lagrangian form, and then convert it into a familiar model and optimization method (in our example, this corresponds to InfoVAE with a specific hyper-parameter choice.)

---

> > ### Comment · AnonReviewer3 · 2017-12-28
> > **thank you for your feedback**
> >
> > Thank you for your feedback.
> >
> > Could you add experiments that optimises the Lagrange multiplier as well? It would help strengthen the paper.

---

> > > ### Author Response · Authors · 2018-01-05
> > > **Solution by Bounding Mutual Information**
> > >
> > > Thank you for your comment. The solution proposed in our paper is to bound the mutual information rather than direct optimization of the Lagrangian multipliers. Direct maximization would lead to maximizing it to infinity for infeasible problems. Our experiments show that bounding the mutual information can solve the problem: as soon as mutual information reaches the preset bound, log likelihood starts to improve.

---

### Official Review · AnonReviewer2 · 2017-11-27
**Good framework for learning generative models, but significance/consequence of the results is unclear**

**Rating:** 5
**Confidence:** 4

**Review:**

Update after rebuttal
==========
Thanks for your response on my questions. The stated usefulness of the method unfortunately do not answer my worry about the significance. It remains unclear to me how much "real" difference the presented results would make to advance the existing work on generative models. Also, the authors did not promised any major changes in the final version in this direction, which is why I have reduced my score.

I do believe that this work could be useful and should be resubmitted. There are two main things to improve. First, the paper need more work on improving the clarity. Second, more work needs to be added to show that the paper will make a real difference to advance/improve existing methods.

==========
Before rebuttal
==========
This paper proposes an optimization problem whose Lagrangian duals contain many existing objective functions for generative models. Using this framework, the paper tries to generalize the optimization problems by defining computationally-tractable family which can be expressed in terms of existing objective functions.

The paper has interesting elements and the results are original. The main issue is that the significance is unclear. The writing in Section 3 is unclear for me, which further made it challenging to understand the consequences of the theorems presented in that section.

Here is a big-picture question that I would like to know answer for. Do the results of sec 3 help us identify a more useful/computationally tractable model than exiting approaches? Clarification on this will help me evaluate the significance of the paper.

I have three main clarification points. First, what is the importance of T1, T2, and T3 classes defined in Def. 7, i.e., why are these classes useful in solving some problems? Second, is the opposite relationship in Theorem 1, 2, and 3 true as well, e.g., is every linear combination of beta-ELBO and VMI is equivalent to a likelihood-based computable-objective of KL info-encoding family? Is the same true for other theorems?

Third, the objective of section 3 is to show that "only some choices of lambda lead to a dual with a tractable equivalent form". Could you rewrite the theorems so that they truly reflect this, rather than stating something which only indirectly imply the main claim of the paper.

Some small comments:
- Eq. 4. It might help to define MI to remind readers.
- After Eq. 7, please add a proof (may be in the Appendix). It is not that straightforward to see this. Also, I suppose you are saying Eq. 3 but with f from Eq. 4.
- Line after Eq. 8, D_i is "one" of the following... Is it always the same D_i for all i or it could be different? Make this more clear to avoid confusion.
- Last line in Para after Eq. 15, "This neutrality corresponds to the observations made in.." It might be useful to add a line explaining that particular "observation"
- Def. 7, the names did not make much sense to me. You can add a line explaining why this name is chosen.
- Def. 8, the last equation is unclear. Does the first equivalence impy the next one?
- Writing in Sec. 3.3 can be improved. e.g., "all linear operations on log prob." is very unclear, "stated computational constraints" which constraints?

---

> ### Author Response · Authors · 2017-12-28
> **Clarification on the Main Concerns**
>
> We thank the reviewers for their time and valuable feedback.
>
> “The main issue is that the significance is unclear.”
>
> Beyond providing an organizational principle for learning objectives (highlighting their information maximization/minimization properties and trade-offs between computational requirements and flexibility) our new perspective is useful for several reasons (Sections 3 and 4):
>
> 1. We are able to characterize **all** learning objectives that can be optimized under given computational constraints (likelihood based optimization; unary likelihood free optimization; binary likelihood free optimization) providing a “closure” result. Even though we do not introduce a new learning objective, we show that (slightly generalized versions) of ten (already known) “base classes” encompass all possible objectives in each category. Therefore, in a certain sense, we show that there do not exist “new” objectives under our stated assumptions on how objectives can be constructed.
>
> 2. We show that several known problems are revealed by the Lagrangian perspective and hold across the entire model family:
>
> a. Correct optimization of the Lagrangian dual requires maximization over the Lagrangian parameters. However, all existing methods use fixed (arbitrarily chosen) Lagrangian parameters. We show failure cases where this does not correctly optimize the primal problem. For example, when the primal objective is information maximization under constraints of distributional consistency, optimization with fixed Lagrangian parameters can maximize mutual information indefinitely without ever encouraging distributional consistency. As a result, data fit (distributional consistency) may even get worse during training (for example, resulting in lower test log likelihood) as mutual information is maximized. We show that this also happens in practice.
>
> b. The Lagrangian perspective allows us to explicitly weight (“price”) different (conflicting) terms in the objective. For example, suppose the input x has more dimensions than the feature space z. Then for the same per-dimension loss, the input space is weighted more than the latent space (because it has more dimensions). We show in the paper that increasing the weight on matching marginals on z can solve the problem and leads to better performance. In general, we can write out the desired preference in Lagrangian form, and then convert it into a familiar model and optimization method (in our example, this corresponds to InfoVAE with a specific hyper-parameter choice.)
>
> “What is the importance of T1, T2, and T3 classes defined in Def. 7, i.e., why are these classes useful in solving some problems?”
>
> It has been observed experimentally that T1 T2 and T3 are increasingly more challenging in terms of optimization stability, sensitivity to hyper-parameters, and outcome of optimization (Arjovsky et al., 2017). In particular, T1 (likelihood based, e.g. VAE) is highly stable and converges quickly, while T2/T3 methods (such as GANs) suffer from issues such as optimization stability, non-convergence. T3 is slightly more challenging than T2 because BiGAN/ALI (Dumoulin et al., 2016a; Donahue et al., 2016) tend to suffer from inaccurate inference.
>
> “Is the opposite relationship in Theorem 1, 2, and 3 true as well, e.g., is every linear combination of beta-ELBO and VMI is equivalent to a likelihood-based computable-objective of KL info-encoding family? Is the same true for other theorems?”
>
>
> Yes the opposite relationship is true as well. The existing objectives enumerated in Theorem 1, 2, 3 are exactly equivalent to T1/T2/T3 computably objectives respectively.
>
> “Third, the objective of section 3 is to show that only some choices of lambda lead to a dual with a tractable equivalent form. Could you rewrite the theorems so that they truly reflect this, rather than stating something which only indirectly imply the main claim of the paper.”
>
> The statement we supported with Theorems 1/2/3 is: only some parameters choices lead to objectives in each computability (T1/T2/T3) classes (easy vs hard to optimize). For example, only parameters choices that correspond to beta-VAE/VMI can have a likelihood-based computable equivalent form. Most objectives cannot be equivalently transformed to become a likelihood based computable objective. We have revised the paper to make the statement more clear.
>
> “Some small comments”
>
> Thank you. We have revised the writing according to the advice.

---

### Official Review · AnonReviewer1 · 2017-11-27
**Not clear what specific insights exist or what problem this solves**

**Rating:** 4
**Confidence:** 4

**Review:**

EDIT: I have read the authors' rebuttals and other reviews. My opinion has not been changed. I recommend the authors significantly revise their work, streamlining the narrative and making clear what problems and solutions they solve. While I enjoy the perspective of unifying various paths, it's unclear what insights come from a simple reorganization. For example, what new objectives come out? Or given this abstraction, what new perspectives or analysis is offered?

---

The authors propose an objective whose Lagrangian dual admits a variety of modern objectives from variational auto-encoders and generative adversarial networks. They describe tradeoffs between flexibility and computation in this objective leading to different approaches. Unfortunately, I'm not sure what specific contributions come out, and the paper seems to meander in derivations and remarks that I didn't understand what the point was.

First, it's not clear what this proposed generalization offers. It's a very nuanced and not insightful construction (eq. 3) and with a specific choice of a weighted sum of mutual informations subject to a combinatorial number of divergence measure constraints, each possibly held in expectation (eq. 5) to satisfy the chosen subclass of VAEs and GANs; and with or without likelihoods (eq. 7). What specific insights come from this that isn't possible without the proposed generalization?

It's also not clear with many GAN algorithms that reasoning with their divergence measure in the limit of infinite capacity discriminators is even meaningful (e.g., Arora et al., 2017; Fedus et al., 2017). It's only true for consistent objectives such as MMD-GANs.

Section 4 seems most pointed in explaining potential insights.  However, it only introduces hyperparameters and possible combinatorial choices with no particular guidance in mind. For example, there are no experiments demonstrating the usefulness of this approach except for a toy mixture of Gaussians and binarized MNIST, explaining what is already known with the beta-VAE and infoGAN. It would be useful if the authors could make the paper overall more coherent and targeted to answer specific problems in the literature rather than try to encompass all of them.

Misc
+ The "feature marginal" is also known as the aggregate posterior (Makhzani et al., 2015) and average encoding distribution (Hoffman and Johnson, 2016); also see Tomczak and Welling (2017).

---

> ### Author Response · Authors · 2017-12-26
> **Clarification on Significance**
>
> We thank the reviewers for their time and valuable feedback.
>
> “It would be useful if the authors could make the paper overall more coherent and targeted to answer specific problems in the literature rather than try to encompass all of them.”
>
> We respectfully disagree. We strongly believe that identifying connections between existing methods and developing general frameworks and theories that encompass as many existing methods as possible is a fundamental scientific goal. Machine learning research is not only about developing new methods and beating benchmarks, but also achieving a deeper understanding of the strengths, weaknesses, and relationships of existing techniques.
>
>
> “What specific insights come from this that isn't possible without the proposed generalization?”
>
> Beyond providing an organizational principle for learning objectives (highlighting their information maximization/minimization properties and trade-offs between computational requirements and flexibility) our new perspective is useful for several reasons:
>
> 1. We are able to characterize **all** learning objectives that can be optimized under given computational constraints (likelihood based optimization; unary likelihood free optimization; binary likelihood free optimization) providing a “closure” result. Even though we do not introduce a new learning objective, we show that (slightly generalized versions) of ten (already known) “base classes” encompass all possible objectives in each category. Therefore, in a certain sense, we show that there do not exist “new” objectives under our stated assumptions on how objectives can be constructed.
>
> 2. We show that several problems are revealed by the Lagrangian perspective and hold across the entire model family:
>
> a. Correct optimization of the Lagrangian dual requires maximization over the Lagrangian parameters. However, all existing methods use fixed (arbitrarily chosen) Lagrangian parameters. We show failure cases where this does not correctly optimize the primal problem. For example, when the primal objective is information maximization under constraints of distributional consistency, optimization with fixed Lagrangian parameters can maximize mutual information indefinitely without ever encouraging distributional consistency. As a result, data fit (distributional consistency) may even get worse during training (for example, resulting in lower test log likelihood) as mutual information is maximized. We show that this also happens in practice.
>
> b. The Lagrangian perspective allows us to explicitly weight (“price”) different (conflicting) terms in the objective. For example, suppose the input x has more dimensions than the feature space z. Then for the same per-dimension loss, the input space is weighted more than the latent space (because it has more dimensions). We show in the paper that increasing the weight on matching marginals on z can solve the problem and leads to better performance. In general, we can write out the desired preference in Lagrangian form, and then convert it into a familiar model and optimization method (in our example, this corresponds to InfoVAE with a specific hyper-parameter choice.)

---

### Decision · Program_Chairs · 2018-01-29
**ICLR 2018 Conference Acceptance Decision**

**Decision:**

Reject

**Comment:**

The paper provides a constrained mutual information objective function whose Lagrangian dual covers several existing generative models. However reviewers are not convinced of the significance or usefulness of the proposed unifying framework (at least from the way results are presented currently in the paper). Authors have not taken any steps towards revising the paper to address these concerns. Improving the presentation to bring out the significance/utility of the proposed unifying framework is needed.